# Halogen-Substituted Triazolethioacetamides as a Potent Skeleton for the Development of Metallo-β-Lactamase Inhibitors

**DOI:** 10.3390/molecules24061174

**Published:** 2019-03-25

**Authors:** Yilin Zhang, Yong Yan, Lufan Liang, Jie Feng, Xuejun Wang, Li Li, Kewu Yang

**Affiliations:** 1College of Biology Pharmacy and Food Engineering, Shangluo University, Shangluo 726000, China; yananan@yeah.net (Y.Y.); lianglouyv@163.com (L.L.); jiefeng_ab123@163.com (J.F.); xuejunwangd@163.com (X.W.); Lily8028@163.com (L.L.); 2Key Laboratory of Synthetic and Natural Functional Molecule Chemistry of Ministry of Education, College of Chemistry and Materials Science, Northwest University, Xi’an 710127, China; kwyang@nwu.edu.cn

**Keywords:** halogen-substitutedtriazolethioacetamides, MβLs, inhibitor

## Abstract

Metallo-β-lactamases (MβLs) are the target enzymes of β-lactam antibiotic resistance, and there are no effective inhibitors against MβLs available for clinic so far. In this study, thirteen halogen-substituted triazolethioacetamides were designed and synthesized as a potent skeleton of MβLs inhibitors. All the compounds displayed inhibitory activity against ImiS with an IC_50_ value range of 0.032–15.64 μM except **7**. The chlorine substituted compounds (**1**, **2** and **3**) inhibited NDM-1 with an IC_50_ value of less than 0.96 μM, and the fluorine substituted **12** and **13** inhibited VIM-2 with IC_50_ values of 38.9 and 2.8 μM, respectively. However, none of the triazolethioacetamides exhibited activity against L1 at inhibitor concentrations of up to 1 mM. Enzyme inhibition kinetics revealed that **9** and **13** are mixed inhibitors for ImiS with *K*_i_ values of 0.074 and 0.27μM using imipenem as the substrate. Docking studies showed that **1** and **9**, which have the highest inhibitory activity against ImiS, fit the binding site of CphA as a replacement of ImiS via stable interactions between the triazole group bridging ASP120 and hydroxyl group bridging ASN233.

## 1. Introduction

β-Lactam antibiotics containing a four membered β-lactam ring are a major class of antibiotics, accounting for over 65% of injectable antibiotics in clinic [1]. However, antibacterial resistance caused by the overuse of antibiotics has become a threat to global public health. The main mechanisms of resistance to β-lactam antibiotics is the expression of β-lactamase from bacteria, which can catalyze the hydrolysis of the C-N bond of the β-lactam ring to deactivate the antibiotics [2,3]. So far, there have been more than 2000 distinct β-lactamases identified [4]. Based on amino acid sequence homology, β-lactamases can be divided into four classes: Classes A, C and D, known as serine-β-lactamases (SBLs), can restore the efficacy of β-lactam antibiotics by a successful clinic combination drug therapy with SBL inhibitors (sulbactam, tazobactam and clavulanic acid) [5,6]; Class B enzymes, known as metallo-β-lactamases (MβLs), display activity by one or two Zn ions in the active sites, which are further divided into subclasses B1-B3 [6,7]. Although a number of promising inhibitor molecules have been reported in recent years, there are no effective inhibitors against MβLs available for clinical therapy [8]. More seriously, new Delphi metallo-β-lactamase 1 (NDM-1), which were first identified from pathogenic bacterium in 2008 and classified as B1 subclass MβLs, greatly aggravates the challenge to treat bacterial infection in clinic because of the ability to hydrolyze almost all β-lactam antibiotics [9,10,11]. Obviously, lack of knowledge about catalytic mechanisms of NDM-1 has delayed the development of clinical inhibitors [12].

In 2016, Mojica et al. reported that the pace of the development of MβLs inhibitors has slowed, which was mainly because a shallow, relatively featureless active site and few scaffolds that can selectively attach to the active site [13]. Mollard et al. reported that thiomandelic acid was capable of potential broad spectrum MβLs inhibitors with *K*_i_ values of 0.09 μM for rthiomandelic acid and 1.28 μM for the S-isomer. Moreover, the thiol group can bind the two zinc ions which is the active site of MβLs [14]. The Weide et al. research group developed more than 200 sulftriazole derivatives, and there were 31 compounds which had inhibitory activity against VIM-2 with *K*_i_ values ranging from 0.01–0.39 μM [15]. The N-methyl mercaptosulfhydrazole compounds substituted containing trifluoromethyl could inhibit MβLs in competing mode, the enzyme activity test showed that the mercapto group was the necessary group to produce the activity [16].

In our group’s preliminary work, we found that the azolylthioacetamides scaffold has potential inhibition for MβLs, and the aryl-substituted thioacetamides containing the triazole group can effectively inhibit ImiS and NDM-1 [17,18,19,20]. In addition, four of the azolylthioacetamides exhibit broad-spectrum inhibitory activity against all three subclasses of MβLs [17]. Based on the research, we want to develop new azolylthioacetamides by conjugation of the triazole and thioacetamides company with halogen to find effective MβLs inhibitors. To further explore the structure-activity relationships of the azolylthioacetamide compounds, thirteen halogen-substituted triazolethio-acetamides were synthesized and eleven of that were novel compounds. Their potential as MβLs inhibitors were evaluated with the enzymes VIM-2, NDM-1, ImiS and L1, which are representatives of the B1, B2 and B3 subclasses of MβLs. Also, molecule docking was adopted to explore the binding method between inhibitors and MβLs.

## 2. Results and Discussion

The thirteen halogen-substituted triazolethioacetamides were designed and synthesized as listed in Figure 1 and the synthetic procedures were available in Scheme 1. Briefly, the crosslink between 2-(5-mercapto-4H-1,2,4-triazol-3-yl) phenol (**s4**) and R-substituted-2-chloro-N-phenyl -acetamide (**N1**-13) gave the corresponding target compounds (**1**–**13**). All compounds were characterized by ^1^H and ^13^C NMR and confirmed by MS.

To test whether these halogen-substituted triazolethioacetamides were inhibitors of the MβLs, three subclasses MβLs B1(VIM-2 and NDM-1), B2 (ImiS), and B3 (L1) were expressed and purified as previously described [21,22,23,24]. The inhibition studies indicated that the halogen-substituted triazolethioacetamides had inhibitory activity against NDM-1, VIM-2 and ImiS, but no activity was observed against L1 at inhibitor concentrations up to 1 mM. The inhibitor concentrations causing a 50% decrease of enzyme activity (IC_50_) was listed in Table 1. As a result, the halogen-substituted triazolethioacetamides **1**–**10** and **13** specifically inhibited ImiS exhibiting an IC_50_ value of less than 1 μM, while **12** showed a slightly higher IC_50_ value of 15.64 μM and **11** nearly had no inhibitory activity with a fluorine at the p-position of anilino group. In particular, **1**–**3** and **9** exhibited higher inhibitory activities against ImiS, and **1** gave the lowest IC_50_ value of 32 nM. On the other hand, **1**–**3** inhibited NDM-1 with an IC_50_ value ranging from 0.17 to 0.96 μM. Meanwhile, **12** and **13** gave the suitable IC_50_ value of 38.9 and 2.8 μM to inhibit VIM-2. This observation suggests that the species, location and amount of halogen substitution can affect the inhibitory activity of triazolethioacetamides in varying degrees. On the whole, triazolethioacetamides containing chlorine (**1**–**8**) increase the inhibitory potency slightly on average compared to those containing fluorine (**9**–**12**), but the presence of trifluoromethoxy (**13**) is beneficial to increase the inhibitory activity. The chlorine or fluorine substitution on the 2-position (**1** and **9**) of anilino group exhibited higher potency than that on the 3-position (**2** and **10**) and 4-position (**3** and **11**). Furthermore, replacement of two hydrogen at the anilino group with chlorine (**4** and **5**) or fluorine (**12**) relatively reduces the inhibitory activity, especially compound **12** with two fluorine at 2,5-position of anilino group. Dramatically, replacing hydrogen at the 2, 4-position with nitro (**6**–**8**) does not change their activities significantly. **2** exhibited the lowest IC_50_ value of 170 nM against NDM-1, which is similar to the data (160 nM) of the best NDM-1 inhibitor among the triazolylthioacetamides that we recently reported [18].

To identify the inhibition mode of the halogen-substituted azolylthioacetamides against MβLs, typical representatives **9** and **13** for ImiS were chosen to determine *K*_i_ values. Lineweaver−Burk plots of ImiS catalyzed hydrolysis of imipenem in the absence and presence of inhibitors are displayed in Figure 2. Compounds **9** and **13** exhibited *K*_i_ values of 0.074 and 0.27 μM, respectively, which are slightly larger than their correlative IC_50_ values. The analysis also demonstrated that all the compounds employed the same partially mixed inhibition type.

Molecular docking analysis was achieved from three comparable conformations (out of 50) docked into corresponding target protein for four representative halogen-substituted triazolethioacetamides. Because there is no high-resolution crystal structure of ImiS available, the very closely related (96% sequence identity) CphA was used instead. The lowest-energy conformations were shown in Figure 3, with the binding energies of −7.6, −7.47, −7.75, −6.99 and −5.87 kcal/mol for the CphA/**1**, CphA/**9**, CphA/**13**, NDM-1/**2** and VIM-2/**13** complexes, respectively.

Docking studies reveal that **1** and **9**, which exhibited the lowest IC_50_ values with ImiS, have similar binding patterns with residue side chains in CphA (Figure 3A,B). The triazole group both interact with ASP120 at distances of less than 3.48 Å, and hydroxyl group of 1 and 9 interact with ASN233 at average distances of 2.11 and 3.25 Å, which may result a 0.13 kcal/mol more favorable binding energy relative to **9**. Compound **13** with trifluoromethoxy group did not bind ASP120 together with Zn(II) ions as **1** and **9** did, but via three different interaction modes to get the lowest bonding energy for CphA with distances between 1.89 and 3.21 Å, the interactions are the amide carbonyl group and ASP264, the trifluoromethoxy group and GLN68, the hydroxyl group interact with ASN233and H196. This may explain why although the docking binding energy of CphA/**1** (−7.6 kcal/mol) and CphA/**9** (−7.47 kcal/mol) complexes are larger than CphA/**13** (−7.75 kcal/mol), **1** and **9** showed more inhibitory potency against CphA. However, the docking mode of VIM-2/**13** complex continued to change, one is the amide carbonyl group form hydrogen bonds with ASN210 and ARG 205, the other is that the interaction between the triazole group and H116, and, in addition, the hydroxyl group that interacts with ASP118. In compound **2**, the hydroxyl group coordinates at Zn(II) ions of NDM-1 in comparable distances of 3.59 Å, the hydroxyl group and triazole group both interact with ASP 124. The docking results reveal that the triazole ring of these compounds plays an important role in the interactions with the close residue side chains, meanwhile there is rarely any direct interaction with active site Zn(II) ions as seen in previous findings, which may be due to the introduction of halogen, nitro and trifluoromethoxy influence the relectrostatic interactions between ligand and target protein.

It was interesting to observe that the compounds had different activity against VIM-2 and NDM-1, even though the active sites of NDM-1and VIM-2 are quite similar. There are two differences between the enzymes which may be significant for the inhibitor binding MβLs: (1) The distance between Zn(II) ion and Asp124 which interacts with the hydroxyl group and the triazole ring of **2** in NDM-1, is bigger in VIM-2 (16.12 Å, PDB code 4NQ2) than in NDM-1 (2.12 Å, PDB code 4EYL). Thus it is impossible to establish effective interaction among the hydroxyl group, Asp124 and Zn(II) ion in VIM-2. (2) VIM-2 has a histidine at position 116 and a aspartic acid at position 118, which are the co-interacting amino acids of active center Zn(II) ion and **13**. However, NDM-1 has a histidine and valine at the same corresponding position, so **13** is difficult to show activity against NDM-1.

Methyl group is the ioisosteric surrogate of chlorine. If the chlorine of compound 1 was replaced by methyl group, the methyl substituted triazolethioacetamides only inhibit NDM-1 with an IC_50_ value of 0.16 μM [18], which is significantly higher than that of compound 1, but no inhibitory activity was observed against ImiS. From Figure 3, it is obvious to observe that the triazolethioacetamides are embedded in the active pocket of ImiS in folding mode, and binded to the NDM-1 active center in a relatively stretched state. The methyl group will increase steric hindrance of inhibitor molecules, and weaken the ability to combine ImiS. Thus, hindrance of substituents can not only change the conformation of triazolethioacetamides, but also affect the interaction between triazolethioacetamides and MβLs.

## 3. Materials and Methods

### 3.1. General Information

General chemicals were purchased from TCI (Tokyo Chemical Industry, Tokyo, Japan) and were used without further purification. All antibiotics used were purchased from Sigma-Aldrich (St. Louis, MO, USA).^1^H NMR and ^13^C NMR spectra were recorded on a NMR spectra were recorded with a Bruker DRX 600 MHz spectrometer (Bruker Daltonics Inc., Billerica, MA, USA). The peaks patterns are indicated as follows: s, singlet; d, doublet; t, triplet; q, quartet; dd, doublet doublet; m, multiplet. The spectra were recorded with TMS as internal standard. Coupling constants (J) were reported in hertz (Hz). Chemical shifts were given in part per million (ppm) on the delta scale. Analytical Thin Layer Chromatography (TLC) was carried out on silica gel F_254_ plates with visualization by ultraviolet radiation. HRMS spectra were recorded on a Bruker MicrOTOF-Q II (Bruker Daltonics Inc., Billerica, MA, USA) mass spectrometer. Inhibition studies were performed on an Agilent-8453 UV-visible spectrometer (Santa Clara, CA, USA).

### 3.2. Synthesis and Characterization

Briefly, R-substituted-2-chloro-N-phenylacetamide (**N1**–**13**) and the intermediate 2-(5-mercapto -4H-1,2,4-triazol-3-yl)phenol (**s4**) were prepared as previously reported [17]. A solution of 2-(5-mercapto-4H-1,2,4-triazol-3-yl) phenol (**s4**) (3 mmol) and NaOH (3.6 mmol) dissolved in H_2_O (15 mL) was added in a 50 mL three-neck round bottomed flask, kept stirring for 30 min. N-substituted-2-chloroacetamides (N1-13) (3 mmol) in hot ethanol (5 mL) was added drop wise to the mix solution, and the slurry was stirred at reflux for 6 h. The resultung solution was cooled and neutralized with HCl (5M) to pH 7.0. The resulting white precipitate (1–13) was filtered off, washed with water (3 × 80 mL), and dried in vacuo. The spectrogram information for the target compounds was shown in the Appendix A.

### 3.3 Determination of IC50 and Ki values

The inhibition studies were carried out on an Agilent UV 8453 spectrophotometer at 25 °C using cefazolin as substrate of CcrA, NDM-1, and L1 and imipenem as substrate of ImiS. Inhibitors **1**–**13** were dissolved in DMSO and then diluted with Tris-HCl, pH 7.0. The substrate concentrations were varied between 25 and 400 μM, and inhibitor concentrations were varied between 125 nM and 1 mΜ. The enzyme and inhibitor were pre-incubated for 30 min before starting the kinetic experiments. The IC_50_ values for all analyzed compounds were calculated based on the kinetic data. The mode of inhibition was determined by generating Lineweaver-Burk plots of the data [23], and the *K*_i_ was determined by replotting the data for slope and intercept versus substrate concentration.

### 3.4. Docking Calculations

Inhibitors were docked into the active sites of NDM-1 (PDB code 4EYL), CphA (PDB code 2QDS), VIM-2 (PDB code 4NQ2). The program AutoDock 4.2 [25] was used for molecule docking analysis. The flexible ligand was docked into each rigid monomeric receptor using a grid box with equal space of 0.375 Å per grid and center of the one or two active-site Zn(II) ions. Fifty conformations were generated for each complex. The rest of the parameters were set at their default values and all docking calculations were performed without constraints. Binding energies were calculated via the Lamarckian genetic algorithm and conformations that constitute each cluster were defined by a root mean square deviation tolerance.

## 4. Conclusions

In summary, we have successfully developed a potent skeleton as MβLs inhibitors, and thirteen halogen-substituted triazolethioacetamides were synthesized and characterized by NMR and MS. Biological activity assays reveal that the triazolethioacetamides have special potency to inhibit ImiS with the lowest IC_50_ value of 32 nM, and compound **1**–**3** with chlorine group display mix inhibition against NDM-1 with a IC_50_ range from 170 to 960 nM. Meanwhile, **12** with two fluorine group and 13 with trifluoromethoxy group show certain inhibitory activity against VIM-2 in vitro. Docking studies reveal that the triazolethioacetamides, which can form stable interactions with the triazole bridging ASP120, and the phenolichydroxyl group interacting with ASN233 in CphA, promote the inhibitory activity against ImiS. The identification of thirteen halogen-substituted triazolethioacetamides which show that a mix mode of inhibition provides potent information for the further development of inhibitors against MβLs.

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
