# Peer review of "Halogen-Substituted Triazolethioacetamides as a Potent Skeleton for the Development of Metallo-β-Lactamase Inhibitors"

_molecules, 2019, doi:10.3390/molecules24061174_

Round 1

Reviewer 1 Report

In the manuscript entitled "Halogen-substituted triazolethioacetamides as a potent skeleton for the development of metallo-b-lactamase inhibitors", Zhang and col. report on a group of compounds that show selectivity against several subclasses of family B lactamases. These triazolethioacetamides are structurally related to some other lactamase inhibitors previously reported by the same group. In the current study, they employ a common synthetic route for the preparation of 13 derivatives, in which all variations have been implemented only in the acetamide moiety, namely the introduction of fluorine and chlorine substitutions at different positions, as well as one trifluoromethoxy at position 4. The results obtained from this battery of compounds show that several of them are potent inhibitors in the low nanomolar range. Additionally, several trends are evident from these data: ortho substitutions seem to be prefered, and chlorine works better than fluorine. However, among the disubstituted derivatives, the authors failed to contemplate the 2,6-dichloro derivative, or any o,o-disubstitution for that matter, despite being the favoured position for introducing a halogen atom. Also, some discussion on how the halogen and its substitution pattern may influence the electron-density of the ring and the basicity/pKa of the acetamide would have been welcome, especially if the authors think the NH is involved in some hydrogen bond interactions. As a last comment on the synthesis of the compound library, the incorporation of a methyl group, as a bioisosteric surrogate of the chlorine atom, would have provided a more complete scenario from which to draw SAR conclusions.

The addition of LB-plots to find out the mode of inhibition of this class of compounds is definitely a plus, but again I believe the authors did not extract all the information available. Especifically, they conclude that the mode of inhibition is mixed, but they do not report the apparent Km value from these plots, or the actual Km value in the absence of inhibitors. This comparison would indicate whether the binding of the inhibitor increases or decreases the affinity of the complex toward the substrate. More importantly, a mixed mode of inhibition tends to point at an allosteric binding pocket for the inhibitor, unlike a competitive mode of inhibition where both substrate and inhibitor fight for the same spot... this should have been taken into consideration when performing docking studies, because obviously the active site cannot accommodate simultaneously both substrate AND inhibitor. Therefore, this incongruity must be first resolved in the discussion before further consideration for publication.

Lastly, use of English and style should be revised.

Author Response

Dear Reviewer,

We have studied the valuable comments from you carefully, and tried our best to revise the manuscript. The point to point responds to the comments are listed as following:

Question 1“…… Additionally, several trends are evident from these data: ortho substitutions seem to be prefered, and chlorine works better than fluorine. However, among the disubstituted derivatives, the authors failed to contemplate the 2,6-dichloro derivative, or any o,o-disubstitution for that matter, despite being the favoured position for introducing a halogen atom.”

Answer: Compound 12 in the manuscript was 2,6-difluoro derivative with an IC50 value of 15.64 μM against ImiS, which is much higher than other compounds. The docking result of complex CphA/12 (Fig 1) showed a binding energy of -7.83 kcal/mol. The energy is lower than others, but the closet approach is 19.4 Å between 12 and the active site of CphA. This is may be the reason that the inhibitory activity of 12 is much lower than other compounds. What's more, in the complex crystal structure of MβLs-triazolethioacetamide we got recently, the proper distance (<5 Å) or effective binding between triazole groups and zinc ions are the key to inhibit MβLs for triazolethioacetamide. A representative crystal structure that has been analyzed was shown in figure 2 (VIM-2-triazolethioacetamide). To sum up, we did not choose to synthesis 2,6-dichloro derivative.

Fig 1 The Docking result of CphA/12

Fig 2 The complex crystal of VIM-2-triazolethioacetamide

Question 2Also, some discussion on how the halogen and its substitution pattern may influence the electron-density of the ring and the basicity/pKa of the acetamide would have been welcome, especially if the authors think the NH is involved in some hydrogen bond interactions.

Answer: Thanks you for the suggestion on the electron-density and basicity/pKa. We really didn't think about doing this through molecular Docking before. In the beginning, we always want to explore the change in conformation, protein folding, the electron-density and basicity/pKa through a difficult process to cultivate complex crystal. We will explore the aspects of electron-density and basicity/pKa by Autodock in the future.

Question 3As a last comment on the synthesis of the compound library, the incorporation of a methyl group, as a bioisosteric surrogate of the chlorine atom, would have provided a more complete scenario from which to draw SAR conclusions.

Answer:We synthesized an o-methyl substituted triazolethioacetamides named 2b in ACS Med. Chem. Lett.2016, 7, 413−417. It only showed a quite better inhibitory activity against NDM-1 with IC50 value of 0.16 μM than compound 1. The discussion was added as follows and labeled from Line 158 to 166. 

“Methyl group is the ioisosteric surrogate of chlorine. If the chlorine of compound 1 was replaced by methyl group, the methyl substituted triazolethioacetamides only inhibit NDM-1 with an IC50 value of 0.16 μM[18], which is significantly higher than that of compound 1, but there is no inhibitory activity was observed against ImiS. From Fig 3, it is observed that the triazolethioacetamides are embedded in the active pocket of ImiS in folding mode, and binded to the NDM-1 active center in a relatively stretched state. The methyl group will increase steric hindrance of inhibitor molecules, and weaken the ability to combine ImiS. Thus hindrance of substituents can not only change the conformation of triazolethioacetamides, but also affect the interaction between triazolethioacetamides and MβLs.”

Question 4The addition of LB-plots to find out the mode of inhibition of this class of compounds is definitely a plus, but again I believe the authors did not extract all the information available. Especifically, they conclude that the mode of inhibition is mixed, but they do not report the apparent Km value from these plots, or the actual Km value in the absence of inhibitors. This comparison would indicate whether the binding of the inhibitor increases or decreases the affinity of the complex toward the substrate. More importantly, a mixed mode of inhibition tends to point at an allosteric binding pocket for the inhibitor, unlike a competitive mode of inhibition where both substrate and inhibitor fight for the same spot... this should have been taken into consideration when performing docking studies, because obviously the active site cannot accommodate simultaneously both substrate and inhibitor. Therefore, this incongruity must be first resolved in the discussion before further consideration for publication.

Answer: (1) Lineweaver−Burk plots were carried out by Sigmaplot 12.0, the artwork of Figure 3 will be showed as an attachment in Sigmaplot 12.0 Notebook format. The figure was cropped for aesthetic reason, and the full picture is shown below.

Fig 3 Full Lineweaver−Burk plot of ImiS catalyzed hydrolysis of imipenem in the absence and presence of 9 (left) and 13 (right).

(2) We are very sorry that all the last batch of enzymes were purified deeply using to prepare the complex crystal, so there are no the same batch enzyme to detect the Km value. Moreover, the new purified protein maybe purified and identified in 5 days and it is exceeded for revision limit time. If the following answers can’t satisfy your concerns, we will attach the relevant data later.

We hope the complex crystal (Fig 2) can explain how the triazolethioacetamide bind to MβLs. In the crystal, the N atom on the triazole ring coordinated with two zinc ions of VIM-2 as replacement of the -OH /H2O, and the carboxyl group forms hydrogen bonds with Asn233, His196, Tyr224, respectively. Compared with the published structures (PDB code: 5lsc), there is no conformational changes in Trp87, but conformational changes were changed in Phe61 side chains. Types and quantities of ligands and metal ions are showed in Table 1.

Table1 Types and quantities of ligands and metal ions in VIM-2- triazolethioacetamide

Ligand

inhibitor

Acetic

acid

Formic

acid

Zn2+

Na+

Cl-

Mg2+

Content

2

2

2

4

4

4

2

Question 5Use of English and style should be revised.

Answer: (1) Line 145: “It was interesting to observe that though NDM-1 and VIM-2 are belong to the same subclass of MβLs……” was revised to “It was interesting to observe that the compounds had different activity against VIM-2 and NDM-1, even though the active sites of NDM-1and VIM-2 are quite similar. There are two differences between the enzymes which maybe significant for the inhibitor binding……

(2) Line 181- “After N-substituted-2-chloroacetamides (N1-13) (3 mmol) dissolved in hot ethanol (5 mL) was added drop wise, the reaction mixture was heated to reflux for 6 h……”was revised to “N-substituted-2-chloroacetamides (N1-13) (3 mmol) in hot ethanol (5 mL) was added drop wise to the mix solution, and the slurry was stirred at reflux for 6 h. The result solution was cooled and neutralized with HCl (5M) to pH 7.0. The resulting white precipitate (1-13) was filtered off, washed with water (3×80 mL), and dried in vacuo. The spectrogram information for the target compounds was shown in the Supplementary Data.

We look forward to hearing from you regarding our submission. We would be glad to respond to any further questions and comments that you may have.

All the authors

Reviewer 2 Report

In the manuscript, authors reported some azolylthioacetamides scaffolds that inhibit Metallo-β-lactamases (MβLs ) activity. The manuscript highlights the potential importance of developing inhibitors against MβLs to overcome antibiotic resistance. This study is well designed and well executed. However, at its current standing the manuscript is incomplete. Therefore, it needs few more experiments to make the paper more interesting.

1.      It is important to see whether any of these compounds potentiates Beta-Lactam antibiotic activity in bacterial. Very least authors should test these compounds against E. coli NDM-1 or VIM2-producing strain. Authors should consider setting up a time course killing curve assay and in vitro mutation assay to test effect of these compounds on potentiating antibiotic activities and overcoming the antibiotic resistance.

2.     Authors can also try to see whether any of the compounds restore antibiotic efficacy against E. coli  NDM-1 or VIM2 producing strain in a murine thigh infection model.

3.       There are some recent articles available on line. Authors need to cite these articles.

4.     In general, English writing needs to improve. In addition, there are some grammatical concerns that need to address. For example,

Line 145: It was interesting to observe that though NDM-1 and VIM-2 are belong to the same subclass of MβLs……

Line 171- After N-substituted-2-chloroacetamides (N1-13) (3 mmol) dissolved in hot ethanol (5 mL) was added drop wise, the reaction mixture was heated to reflux for 6 h……

Author Response

Dear Reviewer,

We have studied the valuable comments from you carefully, and tried our best to revise the manuscript. The point to point responds to the comments are listed as following:

Question 1. It is important to see whether any of these compounds potentiates Beta-Lactam antibiotic activity in bacterial. Very least authors should test these compounds against E.coli NDM-1 or VIM2-producing strain. Authors should consider setting up a time course killing curve assay and in vitro mutation assay to test effect of these compounds on potentiating antibiotic activities and overcoming the antibiotic resistance.

Answer: The ability of the halogen-substituted triazolethioacetamides to restore the antimicrobial activity of antibiotics against bacteria encoding MβLs was investigated by determining the minimum inhibitory concentrations(MIC). Seven antibiotics were choosed to detect the antimicrobial activity of the triazolethio -acetamides, three of which are β-lactam antibiotics. The MIC values of the antibiotics against E.coli-MβLs are showed in Table 1. However, the thirteentriazolethioacetamidesdid not show any antimicrobial activity in the presence and absence of antibiotics. This is maybe most of the drug-resistance bacteriaare Gram-negative pathogens which own two membranes, and the reduction of outer membrane permeability represents amajor mechanism of resistance of Gram-negative pathogens[1]. For the azolethioacetamides with antibacterial activity, our group is carrying out research on reversing the resistance of drug-resistant bacteria and its genetic mechanism by reference from Nature Chemical Biology[2].

Table 1 The MIC values of the antibiotics against E.coli-MβLs (μg/mL)

E.coli-NDM-1

E.coli-VIM-2

E.coli-ImiS

E.coli-L1

cefotaxime   sodium

640

80

5

320

cefazolin   sodium

640

320

20

2560

penicillin

1280

160

40

1280

kanamycin

160

320

320

340

erythromycin

160

160

10

80

tetracycline

80

10

5

20

streptomycin sulphate

80

80

40

20

Reference:

[1] Tan L, Tao YL, Wang T, Zou F, Zhang SH, Kou QH, Niu A, Chen Q, Chu WJ, Chen XY, Wang HD, Yang YS. Discovery of novel pyridone-conjugated monosulfactams as potent and broad-spectrum antibiotics for multidrug-resistant gram-negative infections[J]. Journal of Medicinal Chemistry, 2017, 60 (7): 2669-2684.

[2] Stone LK, Baym M, Lieberman TD, Chait R, Clardy J, Kishony R. Compounds that select against the tetracycline-resistance efflux pump[J]. Nature Chemical Biology, 2016, 12(11):902-904.

Question 2. Authors can also try to see whether any of the compounds restore antibiotic efficacy againstE. Coli-NDM-1 or VIM2 producing strain in a murine thigh infection model.

Answer: We constantlystrive to explore pharmacological and pharmacokinetic effects of MβLsinhibitorsthrough the murine thigh infection model. Unfortunately, hospital in Shaanxi province does not allowed to make murine drug resistance model, but we recently contacted Shanghai institute of medicine which can carry out research in this area, and we believe that great progress will be made in the cooperation this year.The reference literature is as follows.

[1]Tan L, Tao YL, Wang T, Zou F, Zhang SH, Kou QH, Niu A, Chen Q, Chu WJ, Chen XY, Wang HD, Yang YS. Discovery of novel pyridone-conjugated monosulfactams as potent and broad-spectrum antibiotics for multidrug-resistant gram-negative infections [J]. Journal of Medicinal Chemistry, 2017, 60 (7): 2669-2684.

[2] Karuppagounder SS, Alim I, Khim SJ, Bourassa MW, Sleiman SF, John R, Thinnes CC, Yeh TL, Demetriades M, Neitemeier S, Cruz D, Gazaryan I, Killilea DW, Morgenstern L, Xi G, Keep RF, Schallert T, Tappero RV, Zhong J, Cho S, Maxfield FR, Holman TR, Culmsee C, Fong GH, Su Y, Ming GL, Song H, Cave JW, Schofield CJ, Colbourne F, Coppola G, Ratan RR. Therapeutic targeting of oxygen-sensing prolyl hydroxylases abrogates ATF4-dependent neuronal death and improves outcomes after brain hemorrhage in several rodent models[J].Sci Transl Med. 2016, 8(328):328-329.

Question 3. There are some recent articles available on line. Authors need to cite these articles.

Answer:The references are adjusted as follows:

[3] Rizk NA, Kanafani ZA, Tabaja HZ, Kanj SS. Extended infusion ofbeta-lactam antibiotics: optimizing therapy in critically-ill patients in the era of antimicrobialresistance[J]. Expert Rev Anti Infect Ther.2017, 15(7): 645-652.

 [4] Bush K. Past and Present Perspectives on β-Lactamases [J].Antimicrob Agents Chemother, 2018, 62(10): 01076-01118.

[5] Bush K, Bradford PA. Interplay between β-lactamasesand new β-lactamase inhibitors [J]. Nat Rev Microbiol, 2019, doi: 10.1038/s41579-019-0159-8.

 [8] Ju LC, Cheng Z, Fast W, Bonomo RA, Crowder MW. The continuing challenge of metallo-β-lactamase inhibition: mechanism matters [J].Trends Pharmacol Sci. 2018, 39(7): 635-647.

[10] Ali A, Gupta D, Srivastava G, Sharma A, Khan AU. Molecular and computational approaches to understand resistance of New Delhi Metallo β-lactamase variants (NDM-1, NDM-4, NDM-5, NDM-6, NDM-7)-producing strains against carbapenems [J]. J Biomol Struct Dyn, 2018, 1-7.

[25] Forli S, Huey R, Pique ME, Sanner MF, Goodsell DS, Olson AJ. Computational protein-ligand docking and virtual drug screening with the AutoDock suite.Nat Protoc. 2016, 11(5): 905-919.

Question 4. In general, English writing needs to improve. In addition, there are some grammatical concerns that need to address. For example,

Line 145: It was interesting to observe that though NDM-1 and VIM-2 are belong to the same subclass of MβLs……

Line 171- After N-substituted-2-chloroacetamides (N1-13) (3 mmol) dissolved in hot ethanol (5 mL) was added drop wise, the reaction mixture was heated to reflux for 6 h……

Answer: (1) Line 145: “It was interesting to observe that though NDM-1 and VIM-2 are belong to the same subclass of MβLs……” was revised to “It was interesting to observe that the compounds had different activity against VIM-2 and NDM-1, even though the active sites of NDM-1and VIM-2 are quite similar. There are two differences between the enzymes which maybe significant for the inhibitor binding……”

(2) Line 181(now)- “After N-substituted-2-chloroacetamides (N1-13) (3 mmol) dissolved in hot ethanol (5 mL) was added drop wise, the reaction mixture was heated to reflux for 6 h……”was revised to “N-substituted-2-chloroacetamides (N1-13) (3 mmol) in hot ethanol (5 mL) was added drop wise to the mix solution, and the slurry was stirred at reflux for 6 h. The result solution was cooled and neutralized with HCl (5M) to pH 7.0. The resulting white precipitate (1-13) was filtered off, washed with water (3×80 mL), and dried in vacuo. The spectrogram information for the target compounds was shown in the Supplementary Data.

We look forward to hearing from you regarding our submission. We would be glad to respond to any further questions and comments that you may have.

All the authors

Round 2

Reviewer 2 Report

Authors addressed all the comments.